# Dense Interspecies Face Embedding

**Sejong Yang**
Yonsei University
sejong.yang@yonsei.ac.kr

**Subin Jeon**
Yonsei University
subinjeon@yonsei.ac.kr

**Seonghyeon Nam**
York University
snam0331@gmail.com

**Seon Joo Kim**
Yonsei University
seonjookim@yonsei.ac.kr

## Abstract

Dense Interspecies Face Embedding (DIFE) is a new direction for understanding faces of various animals by extracting common features among animal faces including human face. There are three main obstacles for interspecies face understanding: (1) lack of animal data compared to human, (2) ambiguous connection between faces of various animals, and (3) extreme shape and style variance. To cope with the lack of data, we utilize multi-teacher knowledge distillation of CSE and StyleGAN2 requiring no additional data or label. Then we synthesize pseudo pair images through the latent space exploration of StyleGAN2 to find implicit associations between different animal faces. Finally, we introduce the semantic matching loss to overcome the problem of extreme shape differences between species. To quantitatively evaluate our method over possible previous methodologies like unsupervised keypoint detection, we perform interspecies facial keypoint transfer on MAFL and AP-10K. Furthermore, the results of other applications like interspecies face image manipulation and dense keypoint transfer are provided. The code is available at https://github.com/kingsj0405/dife.

## 1 Introduction

Face understanding is a longstanding research topic in computer vision, and there have been extensive studies and applications such as face recognition, facial expression estimation, face photo manipulation, and virtual humans [6, 47, 4, 51, 3]. The vast majority of successful works on faces have focused on human faces due to its practical implication for a variety of applications and the accessibility to a large amount of annotated data. There have been few attempts for understanding faces of other animal species to facilitate more applications such as ecological research, pet identification, and farm automation [14, 34, 12]. However, the task of understanding animal faces is more challenging than that for the human face due to the lack of sufficient data with annotations. In the literature on full-body understanding, joint learning of interspecies data has been studied as a remedy by transferring the knowledge of well-annotated domain such as human to other species [33, 26, 27]. However, the existing approaches are focused on full-body data, therefore their performance on facial data is limited.

In this work, we are interested in the problem of learning the representation of interspecies facial semantics, which is a common representation shared across interspecies facial data. Learning such representation allows us to extend face understanding beyond the human face, as it enables discovering facial semantics of other species without enough annotations through transferring knowledge from well-annotated human data. As many animal species share similar face geometry, the representation can also benefit from learning multiple species jointly. Since sharing photos of animals through social networks has become more popular, learning the interspecies face representation can be used for

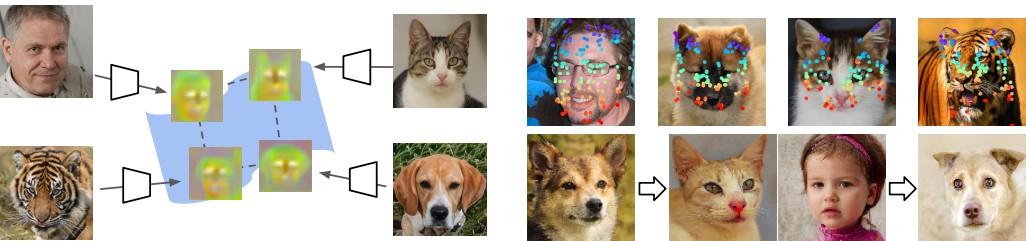

(a) Dense Interspecies Face Embedding (DIFE)          (b) Applications of DIFE

Figure 1: (a) Dense Interspecies Face Embedding (DIFE) is a representation that contains common characteristics of faces of various species embedded to pixels in the feature space. (b) DIFE allows us to find visual correspondence across multiple face domain, so we can transfer facial keypoints between animals or generate different animal faces with the same posture.

various applications. For instance, one can search animal photos using a photo of human face as a query to find animal faces with similar pose. An animal face animation can be synthesized by using that of a human as a template, which can be useful for media production.

To achieve our goal, we propose to learn Dense Interspecies Face Embedding (DIFE) as shown in Fig. 1(a). Our approach is inspired by CSE [27], which was proposed to learn continuous surface embedding (CSE) of full-body shared across interspecies data. While the interspecies embedding in [27] is promising, it is difficult to use it for faces as the embedding in the facial region is too coarse to discriminate facial landmarks accurately. To overcome this problem, we propose a multi-teacher knowledge distillation method for learning DIFE. Our key idea is to enforce a student network to learn a fine-grained interspecies facial embedding from two teacher networks: CSE and StyleGAN2 [19]. Since the CSE has a useful representation of interspecies data, we use it as a teacher to have the student network learn a common embedding space shared in multiple species. For a fine-grained face embedding, we simultaneously exploit StyleGAN2 as the secondary teacher. StyleGAN2 is originally trained to generate high-quality face images [19]. Recent studies have demonstrated that the features of StyleGAN2 have abundant information on facial attributes such as postures and expressions [13, 36, 37]. Thus, the goal of our knowledge distillation is to transfer such information to the student network for learning fine-grained facial semantics for multiple species.

To further improve the performance of DIFE, we propose a data augmentation method to synthesize pseudo-paired images, and the semantic matching loss to enhance the semantic correspondence between images. For the data augmentation, we synthesize images, where the face poses of the synthesized images are similar to that of input images using StyleGAN2. This can be achieved in our framework as DIFE is tied to the latent features of multiple StyleGAN2 networks in different domains. Thus, DIFE of the input image can be used as a face geometry condition for the latent space exploration of StyleGAN2 to generate images for both intra- and inter-domain pairs. In addition, inspired by [39], a semantic matching loss is proposed to use the pseudo-paired data to enforce the DIFE to learn pixel-level fine-grained matching.

We demonstrate the effectiveness of our method on the cross-domain keypoint transfer task using interspecies data such as people, dogs, cats, and wild animals in MAFL [50] and AP-10K [45]. We evaluate our method on both sparse and dense keypoint transfers. In addition, we apply our method for an image editing task of overlaying a virtual object on various animal faces.

Our contribution can be summarized as follows:

- We present a cross-domain face understanding study that considers not only human faces but also the faces of animals. Our approach makes it possible to avoid tedious and expensive animal data collection through domain adaptation, setting the stage for a variety of cross-domain applications.

- We present a novel multi-teacher knowledge distillation paradigm that extracts and combines the information from models with different architectures and data domains. Our framework is designed to learn continuous face embedding across interspecies data from CSE and fine-grained face embedding from StyleGAN2. Combining the knowledge from the two, our method learns dense interspecies face embedding (DIFE).

- We further introduce a method for synthesizing paired data for learning the semantic matching using the latent space exploration of StyleGAN2. With the novel approach, we can refine the representation of our face embedding for semantic correspondence between various instances and species.

## 2 Related Works

### 2.1 Domain-specific Face Understanding

We first review works that aimed to identify the visual correspondence between various instances and scenes within one data category, human or animal. For faces, detecting facial landmarks such as the eyes and the nose has been studied extensively to find correspondences between subjects. Initial works [38, 24, 46, 49, 43, 5] find facial landmarks using facial landmark annotations in a fully supervised way. Regarding animal faces, Yang et al. [44] detected sheep facial landmarks based on 6 annotated keypoints. Although most works show impressive results under extreme conditions, the lack of annotated animal data limits the application compared to human faces.

Recently, there have been active researches [40, 48, 15, 39, 16, 7, 17] on finding landmarks from a collection of images in an unsupervised way, which is applicable to various domains, including human body, face, etc. This stream of works relies on the fact that landmarks are spatially equivariant to the transformation. Some works learn landmarks by comparing warped images with the original ones [40, 48], and others cast the problem into a conditional image generation using video frames depicting the same instances in different poses [15, 16]. The most related work to our method is DVE [39] in that they consider not only equivariance between the same instances but also intra-category variations. All of aforementioned methods can detect landmarks within the same object categories, but the landmarks are not convertible between different object categories (e.g. humans and cats).

### 2.2 Interspecies Understanding

There are studies that aim to discover the relationship between animal species, whose structures are related but geometrically diverse. To the best of our knowledge, [32] is the only work that conducted semantic matching between animal faces. Animal facial landmark detector is trained by transferring the knowledge from the pre-trained human landmark detector. The network is finetuned using horses and sheep data with 5 keypoints annotations. In comparison, we propose an unsupervised approach for interspecies facial semantic matching with the higher granularity of the keypoints.

Closely relevant to our work is the stream of works in [33, 26, 27], as they find the relationship of body poses between animals. Sanakoyeu et al. [33] transfers the knowledge of the human body to the chimpanzee using DensePose [10]. The CSE [26] utilizes the manually annotated 3D meshes of each category and continuous surface embedding to relate several animal categories flexibly. In the following work [27], they utilize the cycle-consistency between meshes and images to further the performance. Although they show impressive results on the complete body, they have difficulty in achieving a fine-level sharpness on faces, making it impossible to discern different parts of faces. While our work also takes the advantage of CSE [27] to detect interspecies correspondences, we improve the accuracy of facial correspondence with the help of the latent features of StyleGAN2 [19] without additional annotations.

### 2.3 StyleGAN2 Utilization

Our work is also related to StyleGAN utilization works in that we use features of pretrained StyleGAN2 [19] to obtain fine-grained facial features. StyleGANs [18, 19] show highly photo-realistic results in the unconditional image generation problem with the power of well disentangled latent spaces. There have been abundant works to exploit the pre-trained StyleGAN for various tasks, including image manipulation [41, 13, 37, 35], 3D reconstruction [30], image segmentation [22, 2], and semantic matching [31].

Because StyleGAN2 has plenty of face geometry many works try to extract the information with knowledge distillation [2, 29, 41]. Labels4Free [2] and Segmentation in style [29] utilizes the feature of StyleGAN2 to train segmentation model in unsupervised manner. In StyleGAN2 distillation [41],

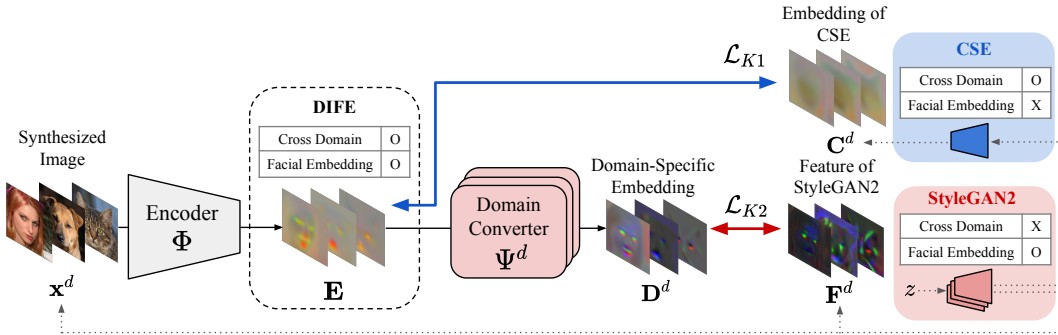

(a) Multi-Teacher Knowledge Distillation

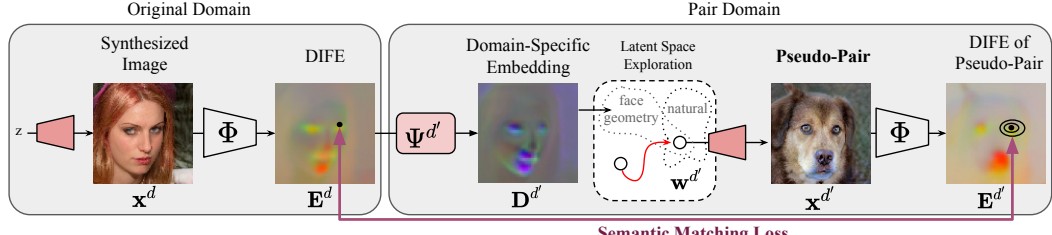

(b) Pseudo-Pair Data Synthesis and Utilization

Figure 2: (a) Our encoder learns the cross domain association and the geometry of face with the responses and the intermediate feature from CSE and multiple StyleGAN2's. The converters for each domain is necessary to generate the domain-specific embeddings which is compatible with the intermediate features of StyleGAN2s. (b) The pseudo pair data of different domains can be synthesized with the domain-specific embedding and the latent exploration of StyleGAN2. The semantic matching loss is utilized to overcome the shape difference between faces of different species.

they generate images and labels with StyleGAN2 generator and face attribute classifier, and image manipulation model is trained with synthesized data. However we can tell no studies have used StyleGAN models learned in different data domains at the same time, while our work learns the relationship between separate StyleGAN2 models, which have different data domains.

## 3 Method

Let $\mathbf{x}^d \in \mathbb{R}^{H \times W \times 3}$ denote an image of a face in a domain $d \in \{0, 1, ..., D-1\}$, where $H$ and $W$ are the height and the width of the image and $D$ denote the number of domains. The goal of our task is to obtain a dense interspecies face embedding $\mathbf{E} \in \mathbb{R}^{H' \times W' \times n}$ of the image $\mathbf{x}^d$ through an encoder $\Phi$, which is described as $\mathbf{E} = \Phi(\mathbf{x}^d)$. Here, $n$ is the embedding dimension for dense interspecies face embedding. To train the encoder $\Phi$ in an unsupervised manner, we propose two training approaches: the multi-teacher knowledge distillation, and the semantic matching through pseudo-paired data synthesis. Since all images and labels are synthesized by the teacher models whose training data are not suitable for our task, the training can be considered as an unsupervised learning. In the following, we describe our dense interspecies face embedding (DIFE) and its training in detail.

### 3.1 Multi-Teacher Knowledge Distillation

Fig. 2(a) illustrates the overview of the multi-teacher knowledge distillation on multiple domains consisting of humans, dogs, and cats. Desirable properties of the interspecies face embedding are two folds. First, the representation is shared by various face images of different domains. Faces in multiple domains should have similar embeddings if their geometry is similar. Second, the embedding contains fine-grained details of facial semantics, so that different parts of a face are distinguishable.

To learn such properties, we first train DIFE by distilling knowledge from CSE [27]. As the representation of CSE has high-quality co-embeddings of diverse animal species, we exploit it to

learn a common space for cross-domain faces. Specifically, we optimize a knowledge distillation loss that minimizes the distance between the DIFE and the CSE embedding of the same input. Formally, the loss is described as

$$\mathcal{L}_{K1}^d = \sum \left\| \mathbf{C}^d - \Phi(\mathbf{x}^d) \right\|_2, \tag{1}$$

where $\mathbf{C}^d$ is the CSE embedding of $\mathbf{x}^d$. Note that $\mathbf{x}^d$ is an image synthesized by StyleGAN2 described as dotted arrow in Fig. 2(a).

Although the above loss enforces our embedding to learn a shared space for multiple domains of faces, the representation of CSE is coarse in the region of the face since it is originally learned to distinguish different parts of a full body. To further learn fine-grained details of faces, we add another knowledge distillation loss using the features of StyleGAN2. Specifically, we exploit the semantic information of faces extracted from a pre-trained StyleGAN2 to enhance the representation of DIFE. Using synthetic images generated by the StyleGAN2 as input, we minimize the distance between the embedding of the input and its corresponding StyleGAN2 feature map, which is formulated as

$$\mathcal{L}_{K2}^d = \sum \left\| \mathbf{F}^d - \mathbf{D}^d \right\|_2, \tag{2}$$

where $\mathbf{F}^d$ is the StyleGAN2 feature map, and $\mathbf{D}^d$ is the domain-specific embedding of $\mathbf{x}^d$ for the domain $d$. Since the dimension of DIFE is different from that of StyleGAN2 and each feature of StyleGAN2s is in the different domain, we convert DIFE to a domain-specific embedding using a domain converter $\Psi^d$, as follows: $\mathbf{D}^d = \Psi^d(\mathbf{E}^d)$. When learning from multiple domains, we optimize $\mathcal{L}_{K1} = \sum_d^D \mathcal{L}_{K1}^d$ and $\mathcal{L}_{K2} = \sum_d^D \mathcal{L}_{K2}^d$ for all of the domains.

## 3.2 Semantic Matching

**Pseudo-paired data synthesis.** Fig. 2(b) shows how to synthesize pseudo-paired data $\mathbf{x}^{d'}$, which has the identical face geometry with the original image $\mathbf{x}^d$. The paired domain $d'$ can be different from the original domain $d$. From a given human face image $\mathbf{x}^d$ generated by StyleGAN2, we first extract DIFE using the encoder $\Phi$. Then, with the domain converter $\Psi^{d'}$, we transform the interspecies embedding into a domain-specific embedding $\mathbf{D}^{d'}$. In Fig. 2(b), we convert DIFE into the dog-specific embedding. Note that the domain-specific embedding lies in the similar vector space with StyleGAN2, since the domain converter is trained by the StyleGAN2 knowledge distillation. Finally, we synthesize a pseudo-paired image $\mathbf{x}^{d'}$ by feeding the domain-specific embedding $\mathbf{D}^{d'}$ to StyleGAN2. Rather than directly feeding the embedding, we introduce a latent space exploration method, which generates more realistic images by adjusting the latent space of the domain-specific embeddings into the natural image manifold.

In the latent space exploration, given a domain-specific embedding $\mathbf{D}^{d'}$, the goal is to find a latent code $\mathbf{w}^{d'} \in \mathcal{W}^{d'}$, which satisfies the following two conditions. Here, $\mathcal{W}^{d'}$ is a latent space of StyleGAN2 trained on the data domain $d'$. First, the pseudo-paired data $\mathbf{x}^{d'}$ generated from the found latent code $\mathbf{w}^{d'}$ should have the same face geometry with the original image $\mathbf{x}^d$. This is achieved by finding $\mathbf{w}^{d'}$ with small difference between the StyleGAN2 feature $\mathbf{F}^{d'}(\mathbf{w}^{d'})$ that is generated from $\mathbf{w}^{d'}$, and the domain-specific embedding $\mathbf{D}^{d'}$. Second, the pseudo-paired data needs to be realistic and natural. To this end, following [1], we take the mean of latent codes $\bar{\mathbf{w}}$, and enforce our latent code to be similar to $\bar{\mathbf{w}}$. $\bar{\mathbf{w}}$ is calculated by sampling 4096 latent codes. With the regularization, the generated image looks natural by locating the latent code in the realistic image manifold. Finally, the latent code $\mathbf{w}^{d'}$ is found by optimizing the following equation:

$$\mathbf{w}^{d'} = \arg\min_{\mathbf{w} \in \mathcal{W}^{d'}} \left[ \left\| \mathbf{F}^{d'}(\mathbf{w}) - \mathbf{D}^{d'} \right\|_2 + \| \mathbf{w} - \bar{\mathbf{w}} \|_2 \right]. \tag{3}$$

Here, the feature map $\mathbf{F}^{d'}(\mathbf{w})$ is extracted from StyleGAN2 by feeding the latent code $\mathbf{w}$ to optimize. After the optimization, the pseudo-paired data $\mathbf{x}^{d'}$ is synthesized by StyleGAN2 with $\mathbf{w}^{d'}$.

**Semantic matching loss.** The purpose of the semantic matching loss is to boost the matching performance of DIFE at pixel level with the synthesized pseudo-paired data. The pixel location $u \in \Omega$ on 2D cartesian coordinate system $\Omega = \{0, 1, ..., H' - 1\} \times \{0, 1, ..., W' - 1\}$ is used to explain

the loss. Theoretically, the pixel embedding $\mathbf{E}_u^d$ at location $u$ in DIFE of the original image has the same semantic meaning with the corresponding pixel embedding $\mathbf{E}_u^{d'}$ of the pseudo-paired image. However, the shape difference between different domains makes it infeasible to exactly match the pixel locations. For example, the eyes of the original image $\mathbf{x}^d$ and the eyes of the pseudo-paired data $\mathbf{x}^{d'}$ in Fig. 2(b) do not perfectly match. To alleviate this issue, we introduce a soft distance measure similar to [39]. Intuitively, we compare the embedding from a pixel on the original image to the weighted embedding of a small region on the pseudo-paired image, as described as the contour in Fig. 2(b). We find the region to compare based on the similarity $\sigma(\mathbf{E}^d, \mathbf{E}^{d'}, u_i, u_j)$ between the pixel embeddings as:

$$\sigma(\mathbf{E}^d, \mathbf{E}^{d'}, u_i, u_j) = \frac{\mathbf{E}_{u_i}^d \cdot \mathbf{E}_{u_j}^{d'}}{\left\| \mathbf{E}_{u_i}^d \right\| \left\| \mathbf{E}_{u_j}^{d'} \right\|}, \tag{4}$$

where $u_i$ and $u_j$ are the pixel coordinates of $\mathbf{E}^d$ and $\mathbf{E}^{d'}$, respectively. Finally, the semantic matching loss $\mathcal{L}_M^{d,d'}$ is defined as follows:

$$\mathcal{L}_M^{d,d'} = \sum_{u_i \in \Omega} \left\| u_i - \sum_{u_j \in \Omega} \sigma(\mathbf{E}^d, \mathbf{E}^{d'}, u_i, u_j) u_j \right\|_2. \tag{5}$$

When learning from multiple domains, we optimize intra semantic matching loss $\mathcal{L}_{M1} = \sum_d^D \mathcal{L}_M^{d,d}$ and inter semantic matching loss $\mathcal{L}_{M2} = \sum_d^D \sum_{d' \neq d}^D \mathcal{L}_M^{d,d'}$ for all permutations of the domains.

### 3.3 Training Objective

We can formulate our final training objective with hyper-parameters as:

$$\mathcal{L} = \lambda_1 \cdot \mathcal{L}_{K1} + \lambda_2 \cdot \mathcal{L}_{K2} + \lambda_3 \cdot \mathcal{L}_{M1} + \lambda_4 \cdot \mathcal{L}_{M2}. \tag{6}$$

The matching losses $\mathcal{L}_{M1}$ and $\mathcal{L}_{M2}$ cannot be applied in the early stage of training, since the domain-specific embedding is different from the features of StyleGAN2s. Therefore, we first impose $\mathcal{L}_{K1}$ and $\mathcal{L}_{K2}$ on the encoder and domain converter, and after initialization, we start to generate pseudo-paired data and utilize semantic matching losses $\mathcal{L}_{M1}$ and $\mathcal{L}_{M2}$.

## 4 Experiments

### 4.1 Interspecies Keypoint Transfer

Given an image in the source domain with keypoints, the goal of interspecies keypoint transfer is to find the corresponding locations of the keypoints on another image in the target domain. We mainly evaluate our method on this task to demonstrate that our embedding is not only a high quality representation of cross-domain faces, but also beneficial for knowledge transfer to unlabeled domains. The task is basically achieved by finding the pixel-level correspondence between the embeddings of the source and the target images. The performance is quantitatively evaluated by calculating the error between the positions of the transferred keypoints and their corresponding ground truth. We further describe mathematical formulations of the evaluation in detail in the supplementary material.

**Datasets.** At the training time, DIFE is trained with synthetic images and their target labels generated by pre-trained CSE and StyleGAN2 networks. We use four datasets for evaluation: MAFL [50], AP-10K [45], WFLW [42] and AnimalWeb [20]. The MAFL is a dataset of human faces that is composed of 20k images. The test set of MAFL excluding the shared images with the training set of CelebA is used (400 images). The AP-10K dataset is composed of 10k images of various animal species which is split into three disjoint subsets, i.e., train, validation, and test sets, with the ratio of 7:1:2 per animal species. As the images contain the full-body of animals, we crop the region of their faces by computing the bounding box using facial landmarks in the dataset. For the evaluation, we make pairs of two different species by randomly selecting images from each domain. As a result, we collect 100 pairs of humans and dogs, 89 pairs of human and cats, 47 pairs of humans and wild animals, and 89 pairs of dogs and cats. The same preprocessing is applied to WFLW and AnimalWeb to evaluate our method on more dense keypoint annotations including the mouth.

Table 1: Quantitative results of interspecies keypoint transfer on MAFL and AP-10K. The pixel error is obtained by calculating the euclidean distance between the ground truth and the transferred keypoints. * indicates that the model is trained by the authors of this paper, as the pre-trained weights are not provided. † indicates that the data domains for the learning and the evaluation do not match.

| Method | Pixel Error | | | | |
|---|---|---|---|---|---|
| | Human+Dog | Human+Cat | Dog+Cat | Human+Dog+Cat | Human+Wild † |
| CSE | 19.00 | 18.15 | 8.00 | 15.20 | 17.69 |
| DVE* | 17.78 | 17.40 | 6.75 | 14.34 | 15.58 |
| CATs | 14.78 | 17.63 | 8.47 | 13.67 | 14.05 |
| Ours(DIFE) | **11.73** | **11.00** | **6.51** | **13.16** | **10.37** |

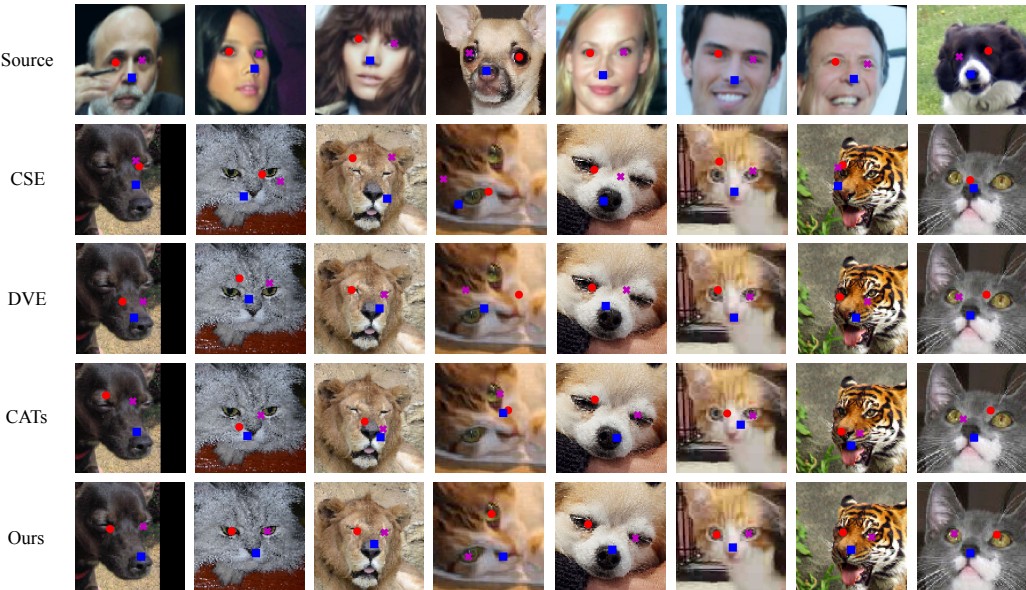

Figure 3: Qualitative results of interspecies keypoint transfer on MAFL and AP-10K.

**Implementation details.** We pre-train CSE with DensePose-COCO [23] and DensePose-LVIS [11], which are the datasets for full-body keypoints of human and animal respectively. StyleGAN2 is pre-trained with FFHQ [19] for human and AFHQ [9] for animals. DIFE encoder uses the hourglass network [28] for a fair comparison with DVE [39], and the domain converter consists of three consecutive $7 \times 7$ conv layers. We use Adam optimizer [21] with the learning rate of $10^{-3}$, batch size of 12, and max training step of 1M with early stopping. The lambda values for each loss functions are fixed to $\lambda_1 = 10^{-2}$, $\lambda_2 = 10^0$, $\lambda_3 = 10^{-2}$, and $\lambda_4 = 10^{-2}$ in all experiments. All experiments are carried out on one NVIDIA Titan XP.

**Baselines.** We compare our method with three baselines: CSE [27], DVE [39] and CATs [8]. Even though CSE is originally learned for full-body embeddings of interspecies data, we compare to the method to demonstrate the effectiveness of our method on face images. DVE is an unsupervised learning approach for learning pixel-level facial embeddings, which is designed for human faces in the original work. For the comparison, we train the method with interspecies data. Both the training set of AP-10K and synthesized images of StyleGAN2 were used, and the latter is selected for better performance. We implemented the baselines using the original source code available on their websites with the default hyperparameters. CATs is the state-of-the-art for finding visual semantic correspondence. We use the pre-trained model from the original paper trained on SPair-71k [25] that contains annotations including the landmarks of dogs, cats, and other animals.

**Quantitative results.** Table 1 presents the quantitative results of CSE, DVE, and DIFE on humans, dogs, cats, and wild animals showing that our method achieves better performance in all data domains.

Table 2: The interspecies keypoint transfer on WFLW [42] and AnimalWeb [20].

|      | Human+Dog | Human+Cat | Dog+Cat | Human+Wild |
|------|-----------|-----------|---------|------------|
| CSE  | 14.43     | *14.95*   | *16.85* | 16.08      |
| DVE  | *14.16*   | 13.28     | 13.40   | *15.74*    |
| CATs | 20.84     | 18.84     | 19.35   | 22.20      |
| Ours | **12.01** | **12.84** | **11.70** | **14.03** |

As can be seen in the table, transferring between human and other animals is more challenging due to the larger shape differences between faces of human and animals. Compared to other baselines, DIFE makes a significant improvement with the help of the semantic loss. The errors in all methods are lower for Dog+Cat, with DIFE performing better than other baselines. We have also tested a challenging case of the keypoint transfer between three domains by training the models at once with human, dogs, and cats, in which DIFE also outperforms other baselines. For the Human+Wild case, we utilized the model that was trained with human+cat, because it was difficult to train with human and wild animals due to the extreme shape and style variance. Since the wild animal domain includes animals that resemble cats, such as tigers, lions, and cheetahs, using our model trained with human and cats performed reasonably, showing that our method is applicable for out-of-domain cases. Our method also outperforms the CATs which indicates the applicability of our method to discovering landmarks in unlabeled animal domains.

Table 2 shows the quantitative result of the interspecies keypoint transfer on WFLW [42] and AnimalWeb [20] with 9 landmarks including the corners of the eye and mouth. Our method shows the best performance compared to previous methods on every domain pair.

**Qualitative results.** Fig. 3 shows some qualitative results of the keypoint transfer. As expected, CSE has difficulty in finely distinguishing different parts of faces as it has limited face prior for animals due to the sparse annotation of DensePose-LVIS. While DVE tends to recognize the eyes and the noses to some extent, it does not properly overcome the shape difference, showing some mismatches. DIFE, with its ability to understand cross-domain association and face embedding, can figure out exact visual correspondence on interspecies faces. More visualizations of the interspecies keypoint transfer including the results on WFLW and AnimalWeb can be found in the appendix.

Fig. 4 visualizes the embeddings of different methods. CSE shows similar values across domains, but less distinction between different parts of a face. On the other hand, StyleGAN2 features exhibit distinctiveness between different parts of a face, but corresponding parts between different domains show a large difference in values, making it difficult to align between different domains. DIFE makes the best of both worlds, showing distinctive values within a face, while having similar values between corresponding face parts between different domains.

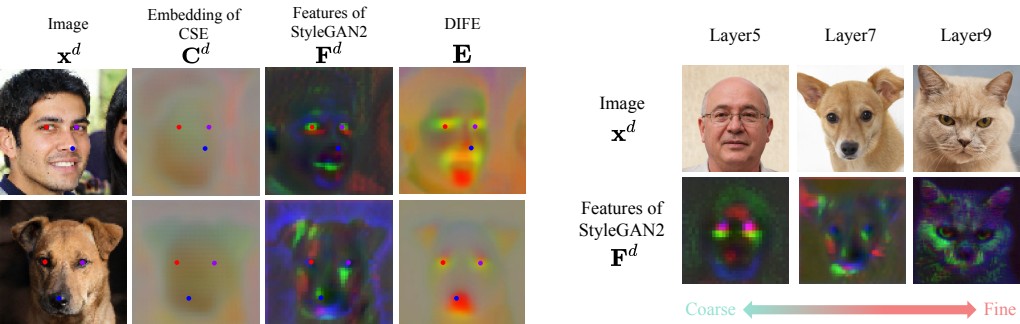

Figure 4: Visualization of embeddings     Figure 5: Visualization of features of StyleGAN2

Table 3: Ablation study on losses

| $\mathcal{L}_{K1}$ | $\mathcal{L}_{K2}$ | $\mathcal{L}_{M1}$ | $\mathcal{L}_{M2}$ | Pixel Error |
|---|---|---|---|---|
| ✓ | | | | 19.33 |
| ✓ | ✓ | | | 13.59 |
| ✓ | ✓ | ✓ | ✓ | **11.73** |
| | ✓ | ✓ | ✓ | 17.45 |
| ✓ | | ✓ | ✓ | - |
| ✓ | ✓ | | ✓ | 13.03 |
| ✓ | ✓ | ✓ | | 12.22 |

Table 4: Quantitative results of choosing layers of StyleGAN2 for the knowledge distillation.

| Layer Number | $n^d$ | Pixel Error |
|---|---|---|
| Layer5 | 512 | 13.80 |
| Layer7 | 512 | **11.73** |
| Layer9 | 256 | 14.78 |

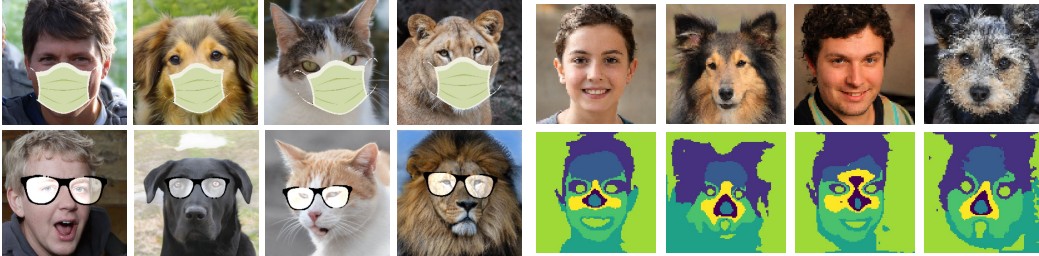

Figure 6: The result of image manipulation          Figure 7: The face parsing results based on DIFE

## 4.2 Ablation Study

**The effect of loss functions.**    Table 3 shows an ablation study to verify the role of each loss. Pixel errors are reduced by adding each loss, and any absence of a loss leads to the performance drop. As we cannot synthesize pseudo-paired data without the knowledge distillation of StyleGAN2, the result for the absence of $\mathcal{L}_{K2}$ is not provided. The absence of $\mathcal{L}_{K1}$ leads to a large performance drop of DIFE, and we find that the role of knowledge distillation of CSE is essential in the early stages of learning. The semantic matching loss also helps to increase the performance by learning more fine visual correspondence across the data domains.

**Which layer in StyleGAN2 is good for the knowledge distillation?**    Table 4 shows the result of selecting a different layer of StyleGAN2 for acquiring the features. Fig. 5 shows that the features from the higher layer is more fine-grained but domain-specific, which makes the domain converter to become confused. Also, because the dimension of the domain-specific embedding follows the dimension of StyleGAN2 features $n^d$, selecting the ninth layer limits the representation power of the domain embedding. On the other hand, the feature from the lower layer is easy for the knowledge distillation with the abundant representation power with large $n^d$, but the information about the face is limited because of its coarseness. With this ablation study, we selected the seventh layer of StyleGAN2 to learn face prior of each data domain.

## 4.3 Applications

**Image manipulation.**    Fig. 6 presents image manipulation examples based on DIFE. Putting objects like glasses and masks requires the positions of face parts, and the result shows that DIFE finds exact locations to put the objects onto the images, even on animal faces.

**Interspecies face parsing**    Fig. 7 shows the results of interspecies face parsing based on DIFE. Following the segmentation in style [29], we use k-means clustering by DIFE for unsupervised face parsing. The eye, nose, mouth, and hairy parts are discovered with a simple method which means DIFE has proper semantic information for the interspecies face.

**Dense keypoint transfer.**    Although DIFE extracts embeddings from all pixels, we cannot accurately evaluate all visual correspondences due to the lack of annotations. However, we can figure out more dense visual correspondence qualitatively by transferring generated grid keypoints . In Fig. 8,

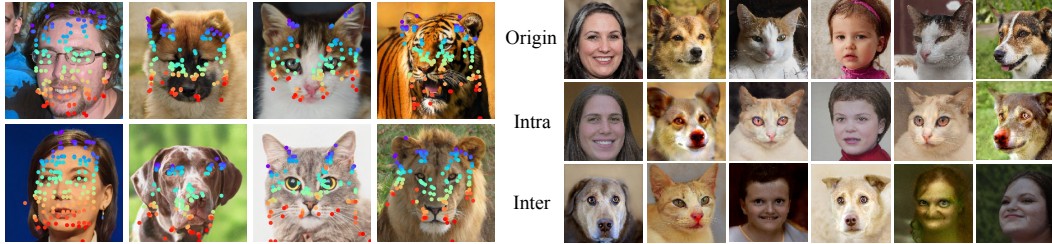

Figure 8: The result of dense keypoint transfer          Figure 9: The examples of pseudo pair data

the result of transferring dense keypoints from a human source to another human, dog, cat, and wild animal targets can be found.

**Pseudo-paired data synthesis.**  Fig. 9 shows examples of pseudo-paired data with various data domains and postures. Given original images, we perform pseudo-paired data synthesis on the intra-domain and the inter-domain. We can see that the generated paired images are in similar poses with the original ones, while having minor texture variations.

## 5   Conclusion

In this paper, we introduced a study for extracting common features from faces of several species, including humans, as a dense embedding. In this process, we presented a mechanism for extracting and learning desired information from separate teacher models, even though the teacher models have different architectures and data domains. Furthermore, we proposed a new data augmentation technique with a latent space exploration of StyleGAN2 and the semantic matching loss to overcome the extreme shape difference between species. Various experiments and applications were offered for the evaluation, and our approach has been shown to be superior to other baselines. For the future research directions, we need to deal with multiple animals more than three domains efficiently, which is not focused on this paper due to the absence of pre-trained StyleGAN2. A research on the generalization method for the out-of-domain data is needed to address the situation where data cannot be obtained or the animal species is unknown. Our method is dependent on the performance of pre-trained CSE and StyleGAN2. Lastly, there are typical face understanding failure cases like occlusion or dark illumination. As the first work to tackle the difficult problem of interspecies face embedding, more follow-up studies are necessary to overcome current limitations. Nevertheless, we believe that this work has set the stage for a new research topic of cross-domain study for interspecies face understanding.

## 6   Broader Impact.

This study aims to understand the interspecies face, which can be potentially extended to various applications. For example, our method can assist to find lost pets, check the health status of farm animals through analyzing the facial expressions. Identifying wild animals for ecological experiments is another expected application of our method. Moreover, the proposed method can potentially be extended to the automatic motion capture, that can be used in the film production or content creation for AR/VR. We note that the studies using DIFE could potentially violate the personal privacy or animal rights. The studies extending DIFE should carefully consider the above potential misuses, and bring positive impacts to the society.

## 7   Acknowledgments

This work was supported by Institute of Information & communications Technology Planning & Evaluation (IITP) grant funded by the Korea government (MSIT) (No.2022-0-00124, Development of Artificial Intelligence Technology for Self-Improving Competency-Aware Learning Capabilities), and No. 2020-0-01361, Artificial Intelligence Graduate School Program (Yonsei University).

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
