| Source | DIFE | Target | DIFE | Source | DIFE | Target | DIFE |
|--------|------|--------|------|--------|------|--------|------|

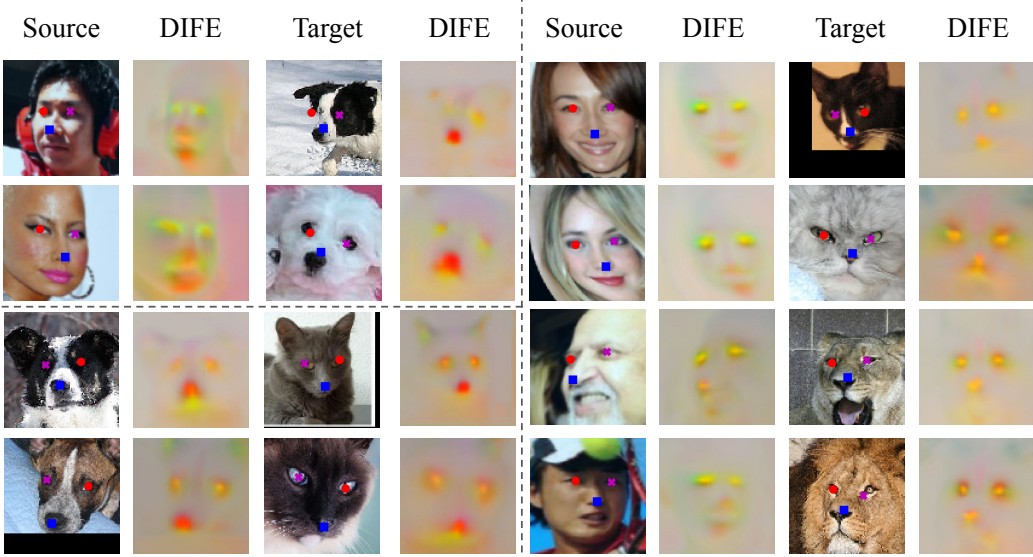

Figure 1: More Qualitative Results with Embedding Visualizations

Table 1: The mean and standard deviation of multiple training.

| DIFE | Pixel Error | | |
|------|-------------|---|---|
| | Human+Dog | Human+Cat | Dog+Cat |
| Mean | 11.73 | 11.00 | 6.51 |
| Standard Deviation | 0.21 | 0.55 | 0.55 |

## A    Metric for Keypoint Transfer

In the keypoint transfer experiment, we calculate the pixel error between the transferred keypoint $\hat{u}_t$ and the ground truth target keypoint $u_t$ to evaluate facial embedding quantitatively. First, source embedding $\mathbf{E}^s$ and target embedding $\mathbf{E}^t$ are extracted from the source image and target image. Then, the ground truth source keypoint $u_s$ is transferred to the target via cosine distance matching of pixel embedding formulated as

$$\sigma(\mathbf{E}^s, \mathbf{E}^t, u_i, u_j) = \frac{\mathbf{E}^s_{u_i} \cdot \mathbf{E}^t_{u_j}}{\left\|\mathbf{E}^s_{u_i}\right\| \left\|\mathbf{E}^t_{u_j}\right\|}, \tag{1}$$

$$\hat{u}_t = \arg\min_{u \in \Omega} \sigma(\mathbf{E}^s, \mathbf{E}^t, u_s, u), \tag{2}$$

with 2D cartesian coordinate system $\Omega = \{0, 1, ..., H' - 1\} \times \{0, 1, ..., W' - 1\}$.

After the keypoint transfer, pixel error $e_p$ is calculated by the euclidean distance between the transferred keypoint and the label of target formulated as following:

$$e_p = \|u_t - \hat{u}_t\|. \tag{3}$$

With lower pixel error, we can assure the learned embedding works well as visual descriptors as matching reliably different face geometry of various species [10].

## B    The reproducibility of our work

For reproducibility of our work, we provide the code blocks in the zipped supplementary file. The codes related to dataset, model, loss, training pipeline and experiment are enclosed. To make sure the training stability of our pipeline, we execute multiple training and interspecies keypoint transfer experiment with respect to the random seed. All values for Section 4 are computed in three randomly seeded training and the statistics for some experiments are shown in Table 1.

Table 2: The human landmark detection results.

| Category | Method | Unsup. | Cross-Domain | MAFL | AFLW$_M$ | AFLW$_R$ | 300W |
|----------|--------|--------|--------------|------|----------|----------|------|
| Supervised learning | TCDCN [13] | X | X | **7.95** | 7.65 | - | 5.54 |
| | MTCNN [12] | X | X | 5.39 | 6.90 | - | - |
| | Wing Loss [3] | X | X | - | - | - | 4.04 |
| Generative modeling based | Deforming AE [9] | O | X | **5.45** | - | - | - |
| | ImGen. [4] | O | X | 2.54 | - | 6.31 | - |
| | ImGEN.++ [5] | O | X | - | - | - | 5.12 |
| Equivariance based | Dense 3D [9] | O | X | **4.02** | **10.99** | **10.14** | **8.23** |
| | DVE [10] | O | X | 2.86 | 7.53 | 6.54 | 4.65 |
| | On Equvariant [1] | O | X | 2.44 | 6.99 | 6.27 | 5.22 |
| | DIFE | O | O | **3.40** | **10.11** | **8.68** | **7.57** |

MAFL     AFLW     300W

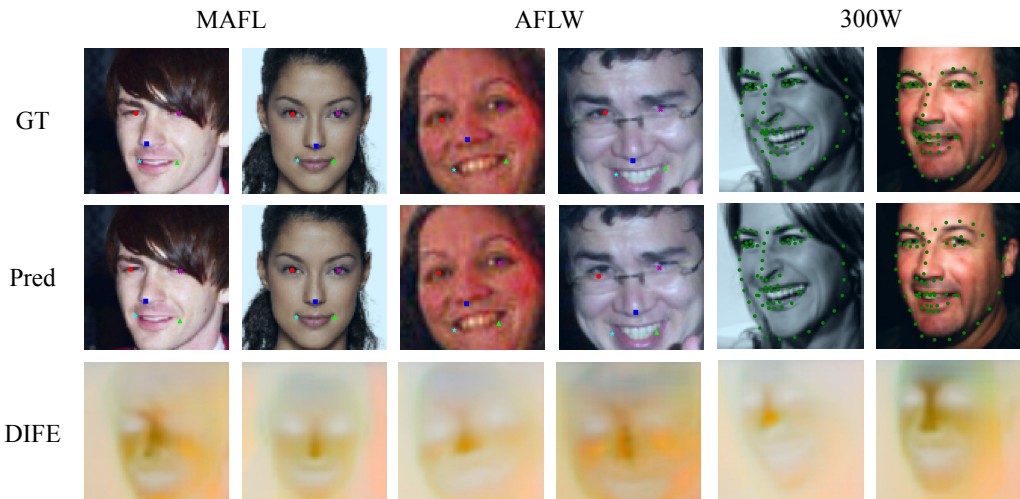

GT  Pred  DIFE

Figure 2: The qualitative results of the human keypoint regression.

## C The human landmark detection

We carry out the keypoint regression experiment setup following DVE [10] and On Equivariant And Invariant [1] to evaluate the robustness and the accuracy of DIFE on intra-species landmark detection. In the experiment, we train a single FC layer with a frozen pre-trained embedder to predict keypoint. The NME values of previous methods are also brought from On Equivariant And Invariant. Even though our embedder is trained on synthesized datasets, not the target dataset, DIFE shows compatible performance with the early study results of each category meaning DIFE is the apposite baseline for cross-domain face understanding. The qualitative results are also provided in Fig. 2.

## D Detailed Explanation for Method

Algorithm 1 shows our detailed training pipeline as pseudo-code. From line 6 to line 18, the pseudo-code for multi-teacher knowledge distillation is provided. At the start of training, latent code $z$ is sampled with normal distribution. The synthesized image $\mathbf{x}^{d1}$, $\mathbf{x}^{d2}$ and the feature of StyleGAN2 $\mathbf{F}^{d1}$, $\mathbf{F}^{d2}$ is generated with the StyleGAN2 $\mathcal{G}^{d1}$, $\mathcal{G}^{d2}$. Here $d1$ and $d2$ means the the domain of instance. The continuous surface embedding $\mathbf{C}^{d1}$, $\mathbf{C}^{d2}$ is extracted with the CSE $\mathcal{E}$. Therfore, DIFE $\mathbf{E}^{d1}$, $\mathbf{E}^{d2}$ and domain-specific embedding $\mathbf{D}^{d1\rightarrow d1}$, $\mathbf{D}^{d2\rightarrow d2}$ are extracted with the encoder $\Phi$ and the domain converter $\Psi^{d1}$, $\Psi^{d2}$. We utilize superscript $d \rightarrow d'$ to express the instance of domain $d'$ is originated from the image of $d$ which is useful to describe pseudo-paired data synthesis later. Finally, CSE knowledge distillation loss $\mathcal{L}_{K1}$ and StyleGAN2 knowledge distillation loss $\mathcal{L}_{K2}$ are calculated with

**Algorithm 1** The Pipeline of Training For Two Domains
___
1: **(0) Prepare DIFE training**
2: Set the learning rate $\eta$
3: Load pre-trained weights For parameters of CSE $\mathcal{E}$, StyleGAN2 $\mathcal{G}^{d1}$, $\mathcal{G}^{d2}$
4: Initialize network parameters $\theta_\Phi$, $\theta_{\Psi^{d1}}$, $\theta_{\Psi^{d2}}$
5: **for** the number of iterations **do**
6:     **(1-1) Synthesize data and labels with teacher models**
7:     Sample latent code $z \sim \mathcal{N}(0, 1)$
8:     $\mathbf{x}^{d1}, \mathbf{F}^{d1} \leftarrow \mathcal{G}^{d1}(z)$
9:     $\mathbf{x}^{d2}, \mathbf{F}^{d2} \leftarrow \mathcal{G}^{d2}(z)$
10:    $\mathbf{C}^{d1}, \mathbf{C}^{d2} \leftarrow \mathcal{E}(\mathbf{x}^{d1}, \mathbf{x}^{d2})$
11:    **(1-2) Extract embedding**
12:    $\mathbf{E}^{d1}, \mathbf{E}^{d2} \leftarrow \Phi(\mathbf{x}^{d1}, \mathbf{x}^{d2})$
13:    $\mathbf{D}^{d1 \to d1} \leftarrow \Psi^{d1}(\mathbf{E}^{d1})$
14:    $\mathbf{D}^{d2 \to d2} \leftarrow \Psi^{d2}(\mathbf{E}^{d2})$
15:    **(1-3) Calculate losses For multi-teacher knowledge distillation**
16:    $\mathcal{L}_{K1} = \mathcal{L}_K(\mathbf{E}^{d1}, \mathbf{C}^{d1}) + \mathcal{L}_K(\mathbf{E}^{d2}, \mathbf{C}^{d2})$
17:    $\mathcal{L}_{K2}^{d1} = \mathcal{L}_K(\mathbf{D}^{d1 \to d1}, \mathbf{F}^{d1})$
18:    $\mathcal{L}_{K2}^{d2} = \mathcal{L}_K(\mathbf{D}^{d2 \to d2}, \mathbf{F}^{d2})$
19:    **if** not initialization period **then**
20:        **(2-1) Intra pseudo-paired data synthesis**
21:        $\mathbf{x}^{d1 \to d1} \leftarrow \mathcal{G}^{d1}(\mathbf{D}^{d1 \to d1})$ with latent space exploration
22:        $\mathbf{x}^{d2 \to d2} \leftarrow \mathcal{G}^{d2}(\mathbf{D}^{d2 \to d2})$ with latent space exploration
23:        **(2-2) Inter pseudo-paired data synthesis**
24:        $\mathbf{D}^{d2 \to d1} \leftarrow \Psi^{d1}(\mathbf{E}^{d2})$
25:        $\mathbf{D}^{d1 \to d2} \leftarrow \Psi^{d2}(\mathbf{E}^{d1})$
26:        $\mathbf{x}^{d2 \to d1} \leftarrow \mathcal{G}^{d1}(\mathbf{D}^{d2 \to d1})$ with latent space exploration
27:        $\mathbf{x}^{d1 \to d2} \leftarrow \mathcal{G}^{d2}(\mathbf{D}^{d1 \to d2})$ with latent space exploration
28:        **(2-3) Extract embedding**
29:        $\mathbf{E}^{d1 \to d1}, \mathbf{E}^{d1 \to d2}, \mathbf{E}^{d2 \to d1}, \mathbf{E}^{d2 \to d2} \leftarrow \Phi(\mathbf{x}^{d1 \to d1}, \mathbf{x}^{d1 \to d2}, \mathbf{x}^{d2 \to d1}, \mathbf{x}^{d2 \to d2})$
30:        **(2-3) Calculate losses For semantic matching**
31:        $\mathcal{L}_{M1} = \mathcal{L}_M(\mathbf{E}^{d1}, \mathbf{E}^{d1 \to d1}) + \mathcal{L}_M(\mathbf{E}^{d2}, \mathbf{E}^{d2 \to d2})$
32:        $\mathcal{L}_{M2} = \mathcal{L}_M(\mathbf{E}^{d1}, \mathbf{E}^{d1 \to d2}) + \mathcal{L}_M(\mathbf{E}^{d2}, \mathbf{E}^{d2 \to d1})$
33:    **else**
34:        $\mathcal{L}_{M1} = 0$
35:        $\mathcal{L}_{M2} = 0$
36:    **end if**
37:    **(3) Update network parameters**
38:    $\theta_\Phi = \theta_\Phi + \eta \nabla_{\theta_\Phi}(\mathcal{L}_{K1} + \mathcal{L}_{K2} + \mathcal{L}_{M1} + \mathcal{L}_{M2})$
39:    $\theta_{\Psi^{d1}} = \theta_{\Psi^{d1}} + \eta \nabla_{\Psi^{d1}}(\mathcal{L}_{K2}^{d1})$
40:    $\theta_{\Psi^{d2}} = \theta_{\Psi^{d2}} + \eta \nabla_{\Psi^{d2}}(\mathcal{L}_{K2}^{d2})$
41: **end for**
___

knowledge distillation loss $\mathcal{L}_K$ formulated as following:

$$\mathcal{L}_K(\mathbf{A}, \mathbf{B}) = \|\mathbf{A} - \mathbf{B}\|_2. \tag{4}$$

The following codes from line 20 to line 32 presents the procedure of the pseudo-paired data synthesis and the semantic matching. The if statement on line 18 means pseudo-paired data is not synthesized in initialization period because the data synthesis requires proper face geometry of the domain-specific embedding. After the initialization period, the intra pseudo-paired data $\mathbf{x}^{d1 \to d1}$, $\mathbf{x}^{d2 \to d2}$ and inter pseudo-paired data $\mathbf{x}^{d1 \to d2}$, $\mathbf{x}^{d2 \to d1}$ are generated with latent space exploration described at Section 3.2. With encoder $\Phi$, each DIFE for pseudo-paired data is extracted. At last semantic matching loss $\mathcal{L}_M$ are utilized to get intra semantic matching loss $\mathcal{L}_{M1}$ and inter semantic matching loss $\mathcal{L}_{M2}$.

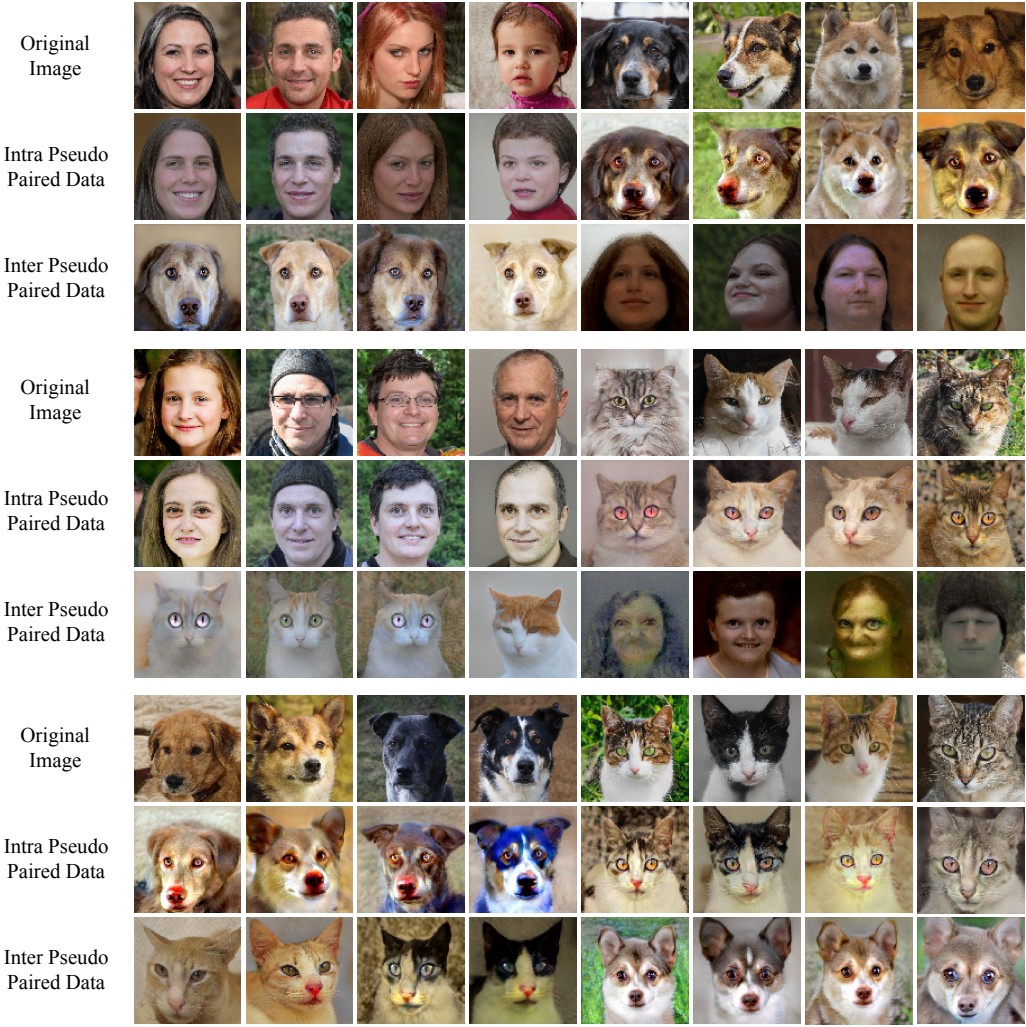

Figure 3: More Examples of Pseudo Paired Data

Here the equation for semantic matching loss with cosine similarity $\sigma$ defined at Appendix B:

$$\mathcal{L}_M(\mathbf{A}, \mathbf{B}) = \sum_{u_i \in \Omega} \left\| u_i - \sum_{u_j \in \Omega} \sigma(\mathbf{A}, \mathbf{B}, u_i, u_j) u_j \right\|_2 . \tag{5}$$

At the end of iteration, the parameters of the encoder $\theta_\Phi$ is updated with all losses and the parameters of the domain converter $\theta_{\Psi^{d1}}$, $\theta_{\Psi^{d2}}$ are updated with StyleGAN2 knowledge distillation loss.

## E  More Examples of Pseudo-Paired Data

We provide more examples of pseudo-paired data on various combinations of original and pair domains in Fig. 3. Each three group of rows shows examples of pseudo-paired data for human+dog, human+cat, and dog+cat. With various pseudo-paired data, we observe the generated image has the same face geometry with the original image. In inter pseudo-paired data, there is a little difference like the location of the ear or the shape of the eye, but the posture of the face is always aligned. By matching them, ambiguous connections between the interspecies faces can be learned in unsupervised manner.

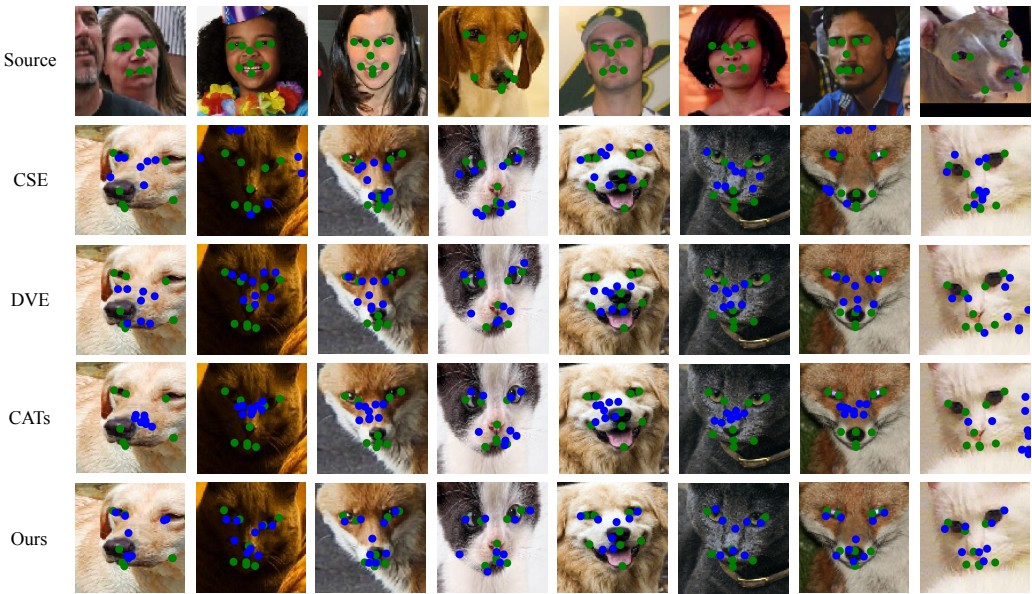

Figure 4: The qualitative results of the interspecies keypoint transfer on WFLW and AnimalWeb. The blue landmarks are predicted and the green landmarks are the ground truth.

## F    Intellectual property rights of assets

The detailed licenses of utilized codes and datasets are as followings:

- Detectron2, DensePose-CSE [8]: The license of code is Aphache 2.0 License which is an open source license.
- StyleGAN, FFHQ [6]: The individual images were published in Flickr by their respective authors under either Creative Commons BY 2.0, Creative Commons BY-NC 2.0, Public Domain Mark 1.0, Public Domain CC0 1.0, or U.S. Government Works license. All of these licenses allow free use, redistribution, and adaptation for non-commercial purposes.
- AFHQ [2]: The dataset is available under Creative Commons BY-NC 4.0 license by NAVER Corporation. There is no problem to use, copy, tranform and build upon the material for non-commercial purposes as long as giving appropriate credit by citing our paper, and indicating if changes were made.
- AP-10K [11]: The dataset is available under Creative Commons BY-NC 4.0 license which is same with AFHQ.
- CelebA [7], MAFL [14]: The name of license is unknown but datasets are available for non-commercial research purposes.