# OpenReview forum: "Dense Interspecies Face Embedding"
_NeurIPS.cc/2022/Conference — NeurIPS 2022 Accept_

### Official Review · Reviewer_tYdV · 2022-07-09

**Rating:** 5
**Confidence:** 3
**Soundness:** 2 fair
**Presentation:** 2 fair
**Contribution:** 2 fair

**Summary:**

This paper addresses the problem of dense interspecies face embedding. To cope with the lack of data, first multi-teacher knowledge distillation of Continuous Surface Embedding (CSE) and StyleGAN is used. Then, pseudo pair images are synthesized through the latent space exploration of StyleGAN2 to find implicit associations between different animal faces. Finally, a semantic matching loss is introduces to overcome the problem of extreme shape differences between species. Interspecies facial keypoint transfer is performed on MAFL and AP-10K. Results for applications in interspecies face image manipulation and dense keypoint transfer are also provided.


**Questions:**

- Authors should convince more about the effective applicability of the method. In particular, I think it would be interesting to see a comparison with a solution that learns face keypoints from animal images rather than transferring them from face images. This would support more the method and is something missing in the paper.


**Limitations:**

Authors reported about the impact of the work but in a too general way about limitations. I think they should have indicated more specifically the limitations possibly also providing failure cases.


**Strengths And Weaknesses:**

Strengths
The main contributions of the paper are:
- A cross-domain face understanding study is presented that considers human faces
as well as faces of animals. The approach domain adaptation to avoid tedious and expensive animal data collection.
- A multi-teacher knowledge distillation paradigm is proposed that extracts and combines the information from models with different architectures and data domains. The proposed framework learns continuous face embedding across interspecies data from CSE and StyleGAN2. With this solution dense interspecies face embedding (DIFE) are learnt.
- A method for synthesizing paired data is proposed for learning the semantic matching using the latent space exploration of StyleGAN2.

Weaknesses
The contribution of the paper is limited. It is more on the application side but it fails to convince about the effective transfer between human an animal faces and application domains.
In particular:
- The baselines used in the comparison of Table 1 are not designed for the same task proposed in this work and have been adapted to it so the comparison is somewhat inconsistent.
- the keypoint transfer is shown only for three keypoints (nose tip and eyes center, but the transfer on the animal face is not particularly accurate)
- the results in figure 6 also are not particularly accurate
- the applications are not particularly convincing

In general, the paper is well written, but there are some errors to check. See for example:
- page 3, line 124: "In StyleGAN2 distillation [? ]," --> missing reference
- page 5, line 177: "is an latent space" --> a latent

============= POST REBUTTAL ================
Authors have answered to most of my questions, so I changed my score to borderline accept. However, I still think the contribution is not very strong.

---

> ### Author Response · Authors · 2022-07-30
> **Reply to Reviewer tYdV by Paper277 Authors**
>
> ### Table 1: The interspecies keypoint transfer on CelebA+AP-10K
> |      | Human+Dog | Human+Cat | Dog+Cat | Human+Wild |
> |------|:---------:|:---------:|:-------:|:----------:|
> |  CSE |   19.00   |   18.15   |   8.00  |    17.69   |
> |  DVE |   17.78   |   *17.40*   |   *6.75*  |    15.58   |
> | CATs |   *14.78*   |   17.63   |   8.47  |    *14.05*   |
> | Ours |   **11.73**   |   **11.00**   |   **6.51**  |    **10.37**   |
> ### Table 2: The interspecies keypoint transfer on WFLW+AnimalWeb
> |      | Human+Dog | Human+Cat | Dog+Cat | Human+Wild |
> |------|:---------:|:---------:|:-------:|:----------:|
> |  CSE |   14.43   |   14.95   |  16.85  |    16.08   |
> |  DVE |   *14.16*   |   *13.28*   |  *13.40*  |    *15.74*   |
> | CATs |   20.84   |   18.84   |  19.35  |    22.20   |
> | Ours |   **12.01**   |   **12.84**   |  **11.70** |    **14.03**   |
> ### Weakness 1 + Weakness 2
> - We understand your concern about the weak connection between the task and the main experiment. As mentioned in Section 4.1 there is no proper dataset or previous works for interspecies face understanding. The interspecies keypoint transfer experiment is an indirect way to evaluate our method quantitatively. We also prepare the best possible previous works as baselines; CSE [23] and DVE [36]. Besides, we carry out the interspecies keypoint transfer on WFLW [R1] and AnimalWeb [R2] datasets in Table 2. The new baseline from the semantic visual correspondence is also added in Table 1 and Table 2.
> - In Table 2,  we report the quantitative result of the interspecies keypoint transfer on WFLW and AnimalWeb with 9 landmarks including the corners of the eye and mouth. The qualitative results of the experiment are updated in Figure 4 of supplementary material. Our method shows the best performance compared to previous methods on every domain pair. We will update the new experiment in the revision.
> ### Weakness 3
> - We understand your concern about the performance of our method as shown in Figure 3. Nevertheless, our method achieves the best performance even in extreme poses (column 4) and out-of-domain (column 7). Because this is the first work for interspecies face understanding, we have more chance to boost the performance in future work. We will update this in the revision.
> ### Weakness 4 + Question
> - We add the interspecies face parsing experiment in Figure 5 of the supplementary material. Following the segmentation in style, we use k-means clustering by DIFE for unsupervised face parsing. The eye, nose, mouth, and hairy parts are discovered with a simple method which means DIFE has proper semantic information for the interspecies face. We will update the result in the revision.
> - The comparison to existing supervised/unsupervised methods is unfair because they focus on intra-domain.
> - Nevertheless, we provide the human keypoint transfer experiment with our method trained on the human-only dataset in Appendix E. Our method show compatible performance to pre-trained DVE which is an unsupervised keypoint detection method for identity-invariant learning inside the human domain.
> - We also add a new baseline CATs [R3] which is the state-of-the-art for finding visual semantic correspondence. We use the pre-trained model from the original paper trained on SPair-71k [R4] that contains annotations including the landmarks of dogs, cats, and other animals. Our method outperforms the CATs both quantitatively and qualitatively, which indicates the applicability of our method to discovering landmarks in unlabeled animal domains. The quantitative results are provided in Table 1 and Table 2. And the qualitative results are provided in Figure 3 and Figure 4 of supplementary materials.
> ### Typo
> - We update the rebuttal revision. Thank you for reading in detail.
> ### Limitations
> - Our method is dependent on the performance of pre-trained CSE and StyleGAN2. There are failure cases on the occlusion, the dark illumination, or the rare species. We will update this in the revision.
> ### References
> - [R1] Wu, Wayne, et al. "Look at boundary: A boundary-aware face alignment algorithm." Proceedings of the IEEE conference on computer vision and pattern recognition. 2018.
> - [R2] Khan, Muhammad Haris, et al. "Animalweb: A large-scale hierarchical dataset of annotated animal faces." Proceedings of the IEEE/CVF Conference on Computer Vision and Pattern Recognition. 2020.
> - [R3] Song, L., Wu, W., Fu, C., Qian, C., Loy, C. C., & He, R. (2021). Pareidolia Face Reenactment. In Proceedings of the IEEE/CVF Conference on Computer Vision and Pattern Recognition (pp. 2236-2245).

---

### Official Review · Reviewer_5viz · 2022-07-10

**Rating:** 6
**Confidence:** 4
**Soundness:** 3 good
**Presentation:** 3 good
**Contribution:** 3 good

**Summary:**

This paper presents an "Interspecies face understanding" task to predict unified spacial features of both human and animal faces.
To solve this problem, it presents a multi-teacher knowledge distillation framework that combines the advantages of different models, i.e. CSE for cross-domain features and StyleGAN2 for facial embedding.
It also explores the latent space of StyleGAN2 to synthesise paired data for semantic matching.

**Questions:**

1. Why not compare on the common face alignment datasets with commonly used metrics?
2. Can this method handle landmarks in mouth and face contour?
3. Are the CSE results in Table 1 actually DIFE trained with L_{K1}? If not, is there any illustration for the similarity between Table 2 and "Human+Dog" column in Table 1?
4. How to guarantee the spatial consistency between the input and output of the Domain Converter? How is this Domain Converter trained?
5. Only synthesised images are used to train the Encoder. If testing on a synthesised dataset, how is the performance? Do you have any experience with this?

**Limitations:**

The authors have discussed the broad impact and potential negative social impact of their work.

**Strengths And Weaknesses:**

Strengths:
1. This paper explores the interspecies facial landmark detection model, which has many potential applications.
2. It is interesting to me to combine two orthogonal models to obtain the cross-domain facial embedding.

Weaknesses:
1. I think the "Interspecies Keypoint Transfer" experiment is not sufficient for the following points:\
  a) Dataset. I think 300W [1] or WFLW [2] for human facial landmarks, and [3] for animal facial landmarks, would be helpful to show the performance of this paper;\
  b) Point Number. Only 3 points are presented as the qualitative results. Can this method handle landmarks in mouth and face contour?\
  c) Comparison. If using the datasets mentioned in 1. a), there can be a comparison with previous supervised/unsupervised methods, which help readers better understand the performance of this paper. The current comparison looks isolated.\
  d) Ablation Study. The ablation study results are partly the same as the "Human+Dog" result in Table 1. So the CSE results in Table 1 are actually DIFE trained with L_{K1}? If so, I think it is improper to call it "CSE" here.
2. There is no constraint to enforce the spatial consistency of the input and output of the Domain Converter.
3. No real images are used when training the Encoder.

Reference

[1] Sagonas, Christos, et al. "300 faces in-the-wild challenge: The first facial landmark localization challenge." Proceedings of the IEEE international conference on computer vision workshops. 2013.

[2] Wu, Wayne, et al. "Look at boundary: A boundary-aware face alignment algorithm." Proceedings of the IEEE conference on computer vision and pattern recognition. 2018.

[3] Khan, Muhammad Haris, et al. "Animalweb: A large-scale hierarchical dataset of annotated animal faces." Proceedings of the IEEE/CVF Conference on Computer Vision and Pattern Recognition. 2020.

---

> ### Author Response · Authors · 2022-07-30
> **Reply to Reviewer 5viz by Paper277 Authors**
>
> ### Table 1: The interspecies keypoint transfer on CelebA+AP-10K
> |      | Human+Dog | Human+Cat | Dog+Cat | Human+Wild |
> |------|:---------:|:---------:|:-------:|:----------:|
> |  CSE |   19.00   |   18.15   |   8.00  |    17.69   |
> |  DVE |   17.78   |   *17.40*   |   *6.75*  |    15.58   |
> | CATs |   *14.78*   |   17.63   |   8.47  |    *14.05*   |
> | Ours |   **11.73**   |   **11.00**   |   **6.51**  |    **10.37**   |
> ### Table 2: The interspecies keypoint transfer on WFLW+AnimalWeb
> |      | Human+Dog | Human+Cat | Dog+Cat | Human+Wild |
> |------|:---------:|:---------:|:-------:|:----------:|
> |  CSE |   14.43   |   14.95   |  16.85  |    16.08   |
> |  DVE |   *14.16*   |   *13.28*   |  *13.40*  |    *15.74*   |
> | CATs |   20.84   |   18.84   |  19.35  |    22.20   |
> | Ours |   **12.01**   |   **12.84**   |  **11.70** |    **14.03**   |
> ### Weakness 1 + Question 1 + Question 2 + Question 3
> - We appreciate the experiment suggestion to evaluate our method from various angles.
> - (1)+(2)+Q2 In Table 2,  we report the quantitative result of the interspecies keypoint transfer on WFLW [R1] and AnimalWeb [R2] with 9 landmarks including the corners of the eye and mouth. The qualitative results of the experiment are updated in Figure 4 of supplementary material. Our method shows the best performance compared to previous methods on every domain pair. We will update the new experiment in the revision. In the case of face contour, we cannot get the quantitative results because of the absence of an animal dataset. However, the qualitative results are provided as dense keypoint transfer results in Figure 8.
> - (3)+Q1 We understand your concern about the isolated results from previous works. However, the direct comparison to existing supervised/unsupervised methods is unfair because they focus on intra-domain. Nevertheless, we provide the human keypoint transfer experiment with our method trained on the human-only dataset in Appendix E. Our method shows compatible performance to pre-trained DVE provided by the original paper.
> - (4)+Q3 Sorry for the typo in Table 2. As you understand, CSE [23] results in Table 1 are the output of pre-trained CSE which is described in Section 4.1(L230-L231). The correct pixel error of DIFE trained with $L_{K1}$ on Table 2 is $19.33$ not $19.00$. We update the rebuttal revision for this.
> ### Weakness 2 + Question 4
> - Because our domain converter just maps the input to the output in the same location with the channel-wise operation, the spatial consistency is preserved.
> ### Weakness 3 + Question 5
> - Although the real image distribution is more helpful to train the model, we cannot use real images as DIFE needs images and corresponding features of StyelGAN2[18] to learn face geometry. Therefore, we use extreme data augmentations like color jittering and thin-plate-spline warping to mimic real image data distribution. We also point out that the model trained with a synthesized dataset achieves the best performance on the real image dataset in Table 1 and Table 2.
> - Q5 Because there is no proper label for synthesized data, we cannot show the quantitative result. However, we conduct dense keypoint transfer on the synthesized images in Figure 8.
> ### References
> - [R1] Wu, Wayne, et al. "Look at boundary: A boundary-aware face alignment algorithm." Proceedings of the IEEE conference on computer vision and pattern recognition. 2018.
> - [R2] Khan, Muhammad Haris, et al. "Animalweb: A large-scale hierarchical dataset of annotated animal faces." Proceedings of the IEEE/CVF Conference on Computer Vision and Pattern Recognition. 2020.

---

> > ### Comment · Reviewer_5viz · 2022-08-06
> > **More experiment results desired to make it a solid work**
> >
> > ## Weakness 1 + Question 1 + Question 2 + Question 3
> >  - It is good to see the experiment results on WFLW and Animalweb, which I think is also helpful to dispel some of other reviewers' concerns.
> >  - **(1)+(2)+Q2.** Since it is hard to evaluate the contour landmarks, I think it would be great to see an intraspecific landmark transfer experiment, which reflects the robustness and accuracy of the method to some extent. One of the methods to conduct this experiment is to make a "mean face", with annotated landmarks, then you map this "mean face", as well as the landmarks, to all images in the testing set, then you are able to figure out the "Normalized Mean Error (NME)" in this dataset. In this way, you can find where you are in the landmark detection task. I did not expect your results to outperform intra-domain results. I just hope to put them in the same metrics. As in many unsupervised methods, they would measure with the same metrics and provide the result of supervised methods for reference.
> >  - **(3)+Q1.** I do not quite understand the setting of the experiment in Appendix E, could you elaborate more about it?
> >
> > ## Weakness 2 + Question 4
> >  - Channel-wise operation does not guarantee the spatial semantic consistency. As shown in the Figure 2 (b), the jaw of the human face DIFE feature is mapped to the dog's mouth in the Domain-Specific Embedding.

---

> > > ### Author Response · Authors · 2022-08-09
> > > **Re-re-reply to Reviewer 5viz by Paper 277 Authors**
> > >
> > > ### Table 3: The human landmark detection
> > > - We bring citations starting with ‘O’ and the NME value of previous works from On Equivariant and Invariant [R1]. For example, ‘O67’ means the 67th reference of the paper. The bold fonts mean lower performance than DIFE.
> > > |          Category          |              Method              | Unsup. | Cross-Domain |   MAFL   |   AFLW_M  |   AFLW_R  |   300W   |
> > > |:--------------------------:|:--------------------------------:|:------:|:------------:|:--------:|:---------:|:---------:|:--------:|
> > > |     Supervised learning    |               TCDCN              |    X   |       X      | **7.95** |    7.65   |     -     |   5.54   |
> > > |                            |            MTCNN [O66]           |    X   |       X      |   5.39   |    6.90   |     -     |     -    |
> > > |                            |          Wing Loss [O14]         |    X   |       X      |     -    |     -     |     -     |   4.04   |
> > > |  Generative modeling based |        Deforming AE [O47]        |    O   |       X      | **5.45** |     -     |     -     |     -    |
> > > |                            |           ImGen. [O28]           |    O   |       X      |   2.54   |     -     |    6.31   |     -    |
> > > |                            |          ImGEN.++ [O29]          |    O   |       X      |     -    |     -     |     -     |   5.12   |
> > > |     Equivariance based     |          Dense 3D [O50]          |    O   |       X      | **4.02** | **10.99** | **10.14** | **8.23** |
> > > |                            |             DVE [O49]            |    O   |       X      |   2.86   |    7.53   |    6.54   |   4.65   |
> > > |                            | On Equvariant And Invariant [R1] |    O   |       X      |   2.44   |    6.99   |    6.27   |   5.22   |
> > > |                            |               DIFE               |    O   |       O      |   3.40   |   10.11   |    8.68   |   7.57   |
> > > ### Weakness 1 + Question 1 + Question 2 + Question 3
> > > - We greatly appreciate your instructive replies toward solid work.
> > > - (1)+(2)+Q2: When we try to choose the mean face and transfer landmarks, the performance was highly dependent on the selected mean face. Instead, we bring the keypoint regression experiment setup following DVE [36] and On Equivariant And Invariant [R1] to evaluate the robustness and the accuracy of DIFE on intra-species landmark detection. In the experiment, we train a single FC layer with a frozen pre-trained embedder to predict keypoint. The NME values of previous methods are also brought from On Equivariant And Invariant [R1]. Even though our embedder is trained on synthesized datasets, not the target dataset, DIFE shows compatible performance with the early study results of each category meaning DIFE is the apposite baseline for cross-domain face understanding. The qualitative results are also provided in Figure 6 of supplementary materials.
> > > - (3)+Q1: In appendix E and Table 2 we conduct the human keypoint transfer experiment with DVE and DIFE trained on only human data. The test dataset is fixed as MAFL [46] and Ap-10k [41] on every row of Table 2. The upper two rows are pre-trained weights from the original paper and the lower two rows are trained by synthesized images from StyleGAN2. The NME values of landmarks are evaluated for same-identity and different-identity following the experiment setup of DVE [36].
> > > ### Weakness 2 + Question 4
> > > - The problem you mentioned is more related to StyleGAN2 latent exploration rather than the spatial consistency of the domain converter. Since we search for a pseudo-paired image (e.g. dog in Fig. 2 (b)) in the manifold of StyleGAN2, it is sometimes difficult to find an image whose face geometry exactly matches that of the input. In this case, we have observed that our latent exploration finds a realistic but unaligned image that some parts are not perfectly aligned with the input. To mitigate the misalignment, we proposed a soft distance measure in a semantic matching loss in Eq (5).
> > > ### References
> > > - [R1] Cheng, Z., Su, J. C., & Maji, S. (2021). On equivariant and invariant learning of object landmark representations. In Proceedings of the IEEE/CVF International Conference on Computer Vision (pp. 9897-9906).

---

### Official Review · Reviewer_pN5V · 2022-07-11

**Rating:** 6
**Confidence:** 4
**Ethics Flag:** Yes
**Soundness:** 2 fair
**Presentation:** 3 good
**Contribution:** 2 fair

**Summary:**

This paper presents a face embedding method to obtain shared semantics between different animal species. The basic idea is to use a knowledge distillation paradigm to extract information from face synthesis (StyleGAN2) and interspecies surface embedding (CSE) models. Specifically, the main encoder is trained so that 1) the embedding is close to the one from CSE and 2) the domain-specific embeddings through converters are close to the features of StyleGAN2. The model also synthesizes pseudo-paired face images using the SyleGAN2 generator to enforce facial semantics correspondence further.

**Questions:**

- As commented above, I would like a more detailed discussion of the training data and pre-training model that this method requires. For example, what images and annotations would be needed to target animals not included in AP-10K or AFHQ?
- Likewise, it would be good to see how much the performance of the pre-training model and the amount of data used for training affects the final performance.
- This is not necessarily a weakness, but it is interesting to note that the eyes are emphasized in the generated images seen in Figure 7 and elsewhere. It would be good to have a discussion on what reason this artifact occurs.

**Ethics Review Area:**

["Privacy and Security (e.g., consent)"]

**Limitations:**

- The above issue of generalization to other animal categories is also mentioned in the text as a limitation.
- As the method deals with faces, sufficient consideration must be given to social impact, which is also mentioned in the text.

**Strengths And Weaknesses:**

## Strengths
- The target task is interesting, and the proposed approach is reasonable to address this goal.
- The performance of the proposed method has been qualitatively and quantitatively verified from various angles, and the improvement from the baseline method is acknowledged.

## Weaknesses
- The proposed method takes the knowledge distillation approach and requires the pre-trained StyleGAN2 and CSE models for training. Compared to previous studies such as DVE [36], it can be seen that the data and annotations required for learning have increased. The performance of the final embedding is expected to be highly dependent on the performance of these two pre-trained models.
- Related to the above, the main contribution of the proposed method is the combination of these two models, and the pseudo-paired data synthesis is not that significant in terms of technical novelty.

---

> ### Author Response · Authors · 2022-07-30
> **Reply to Reviewer pN5V by Paper277 Authors**
>
> ### Weakness 1
> - The performance of DIFE is dependent on the performance of pre-trained CSE [23] and StyleGAN2 [18].
> - It's also true that our method requires more prior compared to DVE [36], but DVE failed to deal with extreme shape and texture variance as shown in the interspecies keypoint transfer. Our approach is  still a more simple way than collecting the dataset for interspecies facial understanding because there are already plenty of unlabeled face images and body-annotated data.
> ### Weakness 2
> - As described in Section 2.3(L126-L128), no studies try to align StyleGAN2 learned in different data domains at the same time. We suggest a new way to utilize the teacher model prior by using the interspecies body model to find the common space for different face generative models. Although the pseudo-paired data synthesis is similar to the latent space exploration method of Image2StyleGAN++ [1], our approach suggests a new way to synthesize paired data on multiple different data domains. In general, our main contribution is a new paradigm to deal with the common space of cross-domain data.
> ### Question 1
> - As shown in the experiment on the human and wild domain, DIFE can handle out-of-domain animals to some extent, meaning that it already found a common space for the cross-domain face.
> If you want to train DIFE from scratch on a new animal with extremely different shapes and textures like fish, you need a 3D reference model for the animal, and three keypoint annotations for the body images to train CSE model. Also, unlabelled face images are required to train the StyleGAN2 model.
> ### Question 2
> - For the performance influence from CSE, our method shows better performance when we change the backbone of CSE to ResNet-101 from ResNet-50 [R1]. The cycle loss for different categories suggested by UniversalMap [24] is also helpful.
> - For the amount of data, our method suffers from overfitting when the number of synthesized images is less than 5k.  However, we observe the amount of training data is not relevant when the number is bigger than 5k. Instead, extreme data augmentations like thin-plate-spline warping and color jittering are more helpful to boost the final performance.
> ### Question 3
> - We only mention the pose of synthesized pseudo-pair data in Section 4.3(L291-L294) because there is no clear conclusion about such a phenomenon. However, we hypothesize a large eye is a more safe way to handle extreme shape variance of eyes. There are more examples of pseudo-paired data in Appendix G.
> ### References
> - [R1] He, K., Zhang, X., Ren, S., & Sun, J. (2016). Deep residual learning for image recognition. In Proceedings of the IEEE conference on computer vision and pattern recognition (pp. 770-778).

---

### Official Review · Reviewer_wX8P · 2022-07-11

**Rating:** 5
**Confidence:** 4
**Soundness:** 3 good
**Presentation:** 3 good
**Contribution:** 2 fair

**Summary:**

The paper presents a new research topic “interspecies face embedding”, which aims to extract common features from faces of several species, including humans, as a dense embedding. The process enables the discovery of other species' facial semantics without enough annotations by transferring knowledge from well-annotated human data. Algorithmically, a multi-teacher knowledge distillation paradigm is introduced to guide unsupervised embedding learning. Experimentally, based on the learned interspecies face embedding, the authors perform interspecies facial keypoint transfer on MAFL and AP-10K datasets.

**Questions:**

Please see the weaknesses.

**Limitations:**

Please see the weaknesses.

**Strengths And Weaknesses:**

Strengths:
+ The problem addressed in the paper is interesting.
+ The proposed method of using multi-teacher knowledge distillation for interspecies face embedding learning is technically sound.
+ The paper is well-organized.


Weaknesses:
Here are some concerns:
1. It’s not clear why the species-specific StyleGAN2 models share geometric consistencies in the learned W latent space.

i) In the pseudo-paired data synthesis stage, it is unclear to me why the pseudo-paired data x^d’ generated from the found latent code should have the same face geometry with the original image x^d (L178-179). In my understanding, there are not necessarily correlations in the learned species-specific W latent spaces.

ii) Is the domain converter species-specific? Did you train separate StyleGAN2 models for different species? Or one universal model for all the animals in AFHQ.

iii) Have you considered training a single StyleGAN2 model by combining FFHQ and AFHQ? Would this make it easier to learn interspecies face embedding? Then the domain converter is not necessary.

2.	The experiment section is quite weak.

i) Qualitative and quantitative results on more keypoints are desirable. The authors only perform keypoint transfer on 3 landmarks. I suggest considering more semantic landmarks, i.e., corners of the mouse etc.

ii) The authors should include more baselines. The current baselines CSE and DVE are not designed for the face. The comparison is thus not convincing to me. I understand there is no literature addressing the same problem. But I think the authors could compare with visual semantic correspondence methods. Although these methods deal with intra-class semantic correspondence, the intra-class deformation of some categories (i.e., car) is more challenging than this interspecies face deformation.

iii) I suggest the authors could add one experiment on interspecies face parsing, which can clearly demonstrate the effectiveness of the proposed method.

iv) More visual results are needed.

3. It’s unclear to me why the pre-trained CSE model is relevant to the face embedding learning and can improve the keypoint transfer performance (based on the results in Tab.2). In my understanding, the CSE model provides Interspecies body priors.

4. Discussions on limitations are needed. Such as, the proposed method may need to train species-specific StyleGAN2 models and domain converters.

---

> ### Author Response · Authors · 2022-07-29
> **Reply to Reviewer wX8P by Paper277 Authors**
>
> ### Table 1: The interspecies keypoint transfer on CelebA+AP-10K
> |      | Human+Dog | Human+Cat | Dog+Cat | Human+Wild |
> |------|:---------:|:---------:|:-------:|:----------:|
> |  CSE |   19.00   |   18.15   |   8.00  |    17.69   |
> |  DVE |   17.78   |   *17.40*   |   *6.75*  |    15.58   |
> | CATs |   *14.78*   |   17.63   |   8.47  |    *14.05*   |
> | Ours |   **11.73**   |   **11.00**   |   **6.51**  |    **10.37**   |
> ### Table 2: The interspecies keypoint transfer on WFLW+AnimalWeb
> |      | Human+Dog | Human+Cat | Dog+Cat | Human+Wild |
> |------|:---------:|:---------:|:-------:|:----------:|
> |  CSE |   14.43   |   14.95   |  16.85  |    16.08   |
> |  DVE |   *14.16*   |   *13.28*   |  *13.40*  |    *15.74*   |
> | CATs |   20.84   |   18.84   |  19.35  |    22.20   |
> | Ours |   **12.01**   |   **12.84**   |  **11.70** |    **14.03**   |
> ### Weakness 1
> - (i) Our key idea to finding an image with the same face geometry of input is to generate StyleGAN2 [18] images conditioned on the facial geometry of the input. This is achieved by minimizing the distance between the embedding of the input and the latent vector of StyleGAN2 as described in Eq. (3). Since the embedding/latent vectors contain the information of face geometry, the StyleGAN2-generated face is optimized to resemble the input. The species-specific latent vector is only used to bridge the domain gap between the input and the StyleGAN2.
> - (ii) The pre-trained StyleGAN2 is trained on each data domain:  FFHQ [18], AFHQ-Dog, and AFHQ-Cat [8]. The shared encoder and species-specific domain-converters are trained in an end-to-end manner with the pre-trained teacher models.
> - (iii) We have tried training StyleGAN2 on the merged dataset but found that the training does not converge. The extreme shape and style variance make StyleGAN2 training unstable. The same reason is behind the absence of pre-trained StyleGAN2 on AFHQ-Wild. In addition, we could not find any publicly available model of StyleGAN2 pre-trained on the merged dataset.
> ### Weakness 2
> - We appreciate the experiment suggestion to evaluate our method from various angles.
> - (i) In Table 2,  we report the quantitative result of the interspecies keypoint transfer on WFLW [R1] and AnimalWeb [R2] with 9 landmarks including the corners of the eye and mouth. The qualitative results of the experiment are updated in Figure 4 of supplementary material. Our method shows the best performance compared to previous methods on every domain pair. The corners of the mouth are hard to match exactly compared to the eye. However, the transferred landmarks by our method lie in the same region, meaning DIFE has an understanding of the semantic face region.
> - (ii) In Table 1 and Table 2, we report the quantitative result of interspecies keypoint transfer by CATs[R3] which is the state-of-the-art method of semantic visual correspondence. The qualitative results of the experiment are updated in Figure 3 and Figure 4 of supplementary material. Because CATs is trained on SPair17k [R4] whose annotations include the annotations on animal bodies, the performance of CATs is limited for the face.
> - (iii) In Figure 5 of the supplementary material we report the results of interspecies face parsing. Following the segmentation in style [26], we apply k-means clustering for DIFE. The eye, nose, mouth, and hairy parts are discovered with a simple method which means DIFE has semantic information for the interspecies face.
> - (iv) We kindly remind you that we have visualized additional qualitative results of representation, keypoint transfer, and pseudo-paired data in the supplementary material. We will also add more results in the revision.
> ### Weakness 3
> - Even though the CSE embedding is mainly a representation of the body, it contains coarse information about faces. It is also observed in experiments shown in Table 1 and Figure 3. Despite the limited quality of face embedding, the CSE [23] provides a good initialization of the common space.
> ### Weakness 4
> - As described above the pre-trained domain converters are not required. However, our method is dependent on the performance of pre-trained CSE and StyleGAN2. There are failure cases on the occlusion, the dark illumination, or the rare species. We will update this in the revision.
> ### References
> - [R1] Wu, Wayne, et al. "Look at boundary: A boundary-aware face alignment algorithm." CVPR. 2018.
> - [R2] Khan, M. H., McDonagh, J., Khan, S., Shahabuddin, M., Arora, A., Khan, F. S., ... & Tzimiropoulos, G. Animalweb: A large-scale hierarchical dataset of annotated animal faces. CVPR. 2020.
> - [R3] Cho, S., Hong, S., Jeon, S., Lee, Y., Sohn, K., & Kim, S.. Cats: Cost aggregation transformers for visual correspondence. NeuIPS. 2021.
> - [R4] Min, J., Lee, J., Ponce, J., & Cho, M. Spair-71k: A large-scale benchmark for semantic correspondence. arXiv preprint arXiv:1908.10543. 2019.

---

> > ### Comment · Reviewer_wX8P · 2022-08-10
> > **Acknoledgement of rebuttal**
> >
> > Thanks for your answers. They addressed most of my concerns pre-rebuttal. I will update my score.

---

### Review · Ethics_Reviewer_Bwn5 · 2022-07-23

**Recommendation:**

The authors would make this a stronger paper by testing for bias or at minimum sharing the limitation of showing primarily white faces.
Additionally, they should acknowledge that facial expressions don't explicitly transfer or share emotions between species. A helpful paper could be "Facial Expression in Nonhuman Animals", but generally more psychology understanding would be more helpful here.

**Ethical Issues:**

Yes

**Ethics Review:**

- Representational Biases in human dataset
- Preventing misuse is acknowledged but could go deeper.
- Psychologically expressions can differ by species as to what that emotion means.
- Animal rights

---

### Review · Ethics_Reviewer_ZjLK · 2022-08-07

**Recommendation:**

Yes, these concerns can be addressed in the current version of the paper, but (1) adding a reflection in the Broader Impacts section and (2) explaining the demographic representativeness of the training corpora (for humans) and quantifying performance for different racial / ethnic / gender groups.

**Ethical Issues:**

Yes

**Ethics Review:**

This paper is about understanding the faces of animals (e.g., identifying key points such as location of the eyes, nose, or mouth) by extracting common features among the faces of other animals and of humans. This use case raises several ethical concerns:

1) Data protection / privacy: The data rights of humans whose facial features are being mapped to the facial features of animals.  It may well be the case that a human would object to such use of their data.

2) Racial / ethnic / gender bias: Performance of the proposed method is not evaluated specifically on human faces of different racial / ethnic groups and of different genders.  It may well be the case that the method works better or less well for mapping the faces of members of specific demographic groups to specific animal species.   The outrageous example that comes to mind is when an AI built by Facebook labelled videos of Black men with "Primates" (e.g.,  https://www.nytimes.com/2021/09/03/technology/facebook-ai-race-primates.html).  While the method described here is not used for image or video labeling, the issue of demographic bias deserves to be further explored experimentally, reflected upon, and acknowledged in the writing.

---

### Meta-Review · Area_Chair_LD3r · 2022-08-25

**Recommendation:** Accept
**Confidence:** Less certain

**Metareview:**

This paper uses knowledge distillation to transfer learn facial embeddings across humans and animals. Helps when sufficient data for learning embeddings from animal faces is not available. An interesting application of standard concepts from domain adaptation, knowledge distillation, etc. While preparing the final paper the authors may highlight the novelty in this work. The paper is acceptable.

**Award:**

No

---

### Decision · Program_Chairs · 2022-09-14

Accept